The effects of Solidago canadensis water extracts on maize seedling growth in association with the biomass allocation pattern

Ye Xiao qi
Meng Jin liu
Wu Ming hangzhoubay@126.com
Research Station of Hangzhou Bay Wetland Ecosystems, National Forestry Bureau/Institute of Subtropical Forestry, Chinese Academy of Forestry , Hangzhou , P.R.China
Berninger Frank
Electronic publication date: 2019 Mar 12
Publication date: 2019
Volume: 7
Electronic Location ID: e6564
Received 2018 Jul 3; Accepted 2019 Feb 2
Copyright: © 2019 Ye et al.
Copyright year: 2019
Copyright holder: Ye et al.
License: This is an open access article distributed under the terms of the Creative Commons Attribution License, which permits unrestricted use, distribution, reproduction and adaptation in any medium and for any purpose provided that it is properly attributed. For attribution, the original author(s), title, publication source (PeerJ) and either DOI or URL of the article must be cited.
License URL: https://creativecommons.org/licenses/by/4.0/

Keywords: Water extract, Photosynthetic capacity, Allelopathic effects, Invasive species, Leaf area ratio

Funding: Special Funds for Basic Science Research of Central Public Research Institutes CAFYBB2016SY010 National Natural Science Foundation of China 31770578 Natural Science Foundation of Zhejiang Province LY17C030002 This work was supported by the Special Funds for Basic Science Research of Central Public Research Institutes (CAFYBB2016SY010), the National Natural Science Foundation of China (31770578) and the Natural Science Foundation of Zhejiang Province (LY17C030002). The funders had no role in study design, data collection and analysis, decision to publish, or preparation of the manuscript.

==============================
Background

Solidago canadensis L. is an aggressive exotic plant species in China that has potential allelopathic effects on competing plant species. Effects of hormesis are frequently observed in studies of allelopathy; however, the mechanisms of such effects need to be elucidated. Allelopathic compounds may affect the growth of recipient plants via alteration of biomass allocation patterns or photosynthetic capacity. The aim of this study was to determine how water extracts from S. canadensis affected the shoot and root growth of recipient plants and whether the underlying mechanism was related to the biomass allocation pattern or photosynthetic gas exchange capacity.

Methods

The water extracts from S. canadensis shoots at 12 different concentrations in the range of 0–0.25 g/ml were applied thrice in 9 days to maize seedlings cultivated in silica sand. The growth (shoot height, leaf length and area and root length) and biomass accumulation and allocation (specific leaf area (SLA), leaf area ratio (LAR) and leaf mass ratio (LMR)) were compared among maize seedlings exposed to different treatment concentrations. Gas exchange (photosynthetic light response curve) was measured and compared among maize seedlings exposed to three concentrations of water extract (0, 0.0125 and 0.2 g/ml) before and after the first application, and seedling growth was measured after the third and final application.

Results

The growth of seedlings (shoot height, leaf length and area and root length) was promoted at concentrations below 0.125 g/ml and inhibited at concentrations above this level (P < 0.05). The pattern of change in biomass accumulation and allocation was similar to that of shoot growth, but biomass accumulation and allocation was not significantly affected by the water extract treatments (P > 0.05). The water extract treatments did not significantly affect the photosynthetic capacity (P > 0.05), but the dark respiration rate was higher in the low-dose treatment than that in the high-dose treatment. Shoot height was positively correlated with the biomass allocation indicators SLA and LAR (P < 0.05) but not with LMR (P > 0.05).

Conclusions

The results suggested that the effects of the water extracts from S. canadensis were highly dependent on the concentration, with the growth of maize seedlings promoted at low concentrations of water extracts. The effects of the water extracts on the growth of maize seedlings were mainly due to the effects on the LAR, the allocation to leaf area growth, whereas the effects of the water extracts on leaf gas exchange capacity cannot explain variation of seedling growth. Thus, the stimulation of plant growth was very likely due to increased biomass allocation towards the shoot.

Introduction

Solidago canadensis L. (Asteraceae), which originates from North America, has successfully invaded southeastern China. This species usually forms large monocultures and causes substantial yield loss of crop plants (Liu et al., 2010). Although how the invasiveness of this species develops is unknown, the hypothesis that allelopathy may contribute to the success of the species is supported by some existing evidence (Abhilasha et al., 2008; Yuan et al., 2013). The rhizome extract of S. canadensis imposed effects of hormesis on both the growth and physiological activity of lettuce seedlings in a laboratory experiment (Zhang et al., 2012). Therefore, whether the allelochemicals of S. canadensis increase or decrease crop yields may depend on their concentration in soils, which should be considered when explaining the interaction of S. canadensis with native species or crop plants.

Hormesis refers to the stimulation of organism performance that occurs at low levels of exposure to agents that are harmful or toxic at high levels of exposure (Forbes, 2000; Calabrese & Baldwin, 2001). The hormetic effects of herbicides on plant growth have been observed many times (Cedergreen, 2008a), while more recently, the hormetic effects of phytotoxins have received attention due to their close association with exotic plant invasion (Prithiviraj et al., 2007; Zhang et al., 2012).

Many mechanisms to explain hormetic effects have been proposed (Prithiviraj et al., 2007; Duke et al., 2006). The induction of defense mechanisms induced by free radicals of oxygen can lead to increased growth at low doses of phytotoxic chemicals (Kovalchuk et al., 2003). For example, a low dose of (±)-catechin, which is produced by the invasive weed Centaurea maculosa, induced moderate increases in reactive oxygen species in meristems and much greater biomass accumulation (Prithiviraj et al., 2007). Moreover, reactive oxygen species have proven to be essential for cell elongation in plants (Rodríguez, Grunberg & Taleisnik, 2002). Some chemicals that can affect plant secondary metabolism are associated with the synthesis of cell wall fibers at low doses (Duke et al., 2006). For example, glyphosate inhibits the shikimate pathway, the source of lignin precursors, and might preferentially inhibit lignin synthesis at low, nonherbicidal doses, making cell walls more elastic for longer periods during development (Duke et al., 2006). The roles of photosynthetic capacity in explaining growth promotion effects by low-dose chemicals have also been carefully examined. Compared with untreated barley plants (Cedergreen, 2008b), when sprayed with low doses of glyphosate, barley plants had a higher relative growth rate (RGR) that was partly attributed to the increased photosynthesis rate (Cedergreen & Olesen, 2010). Increased photosynthesis rate was also found in the promotion effects of cadmium, a heavy metal, on the growth of several plant species (Jia et al., 2015; Pereira et al., 2016). Nevertheless, enhanced photosynthesis was not observed when growth was stimulated in cucumber plants treated with several herbicides (Wiedman & Appleby, 1972) or in rice plants treated with microcystins (Liang & Wang, 2015). These inconsistent results suggest that other critical responses explain the dose effects on plant growth, such as biomass allocation patterns. Considering that the plant RGR consists of a morphological component, the leaf area ratio (LAR), and a physiological component, the net assimilation rate (Poorter, 1990), an increase in the RGR under a low dose of a hormetic substance is not necessarily caused by an increase in the photosynthesis rate but can also be due to an increase in the allocation to leaf area. Indeed, some hypotheses state that the stimulatory growth at low concentrations is due to altered resource allocation between shoots and roots (Duke et al., 2006). Therefore, to best explain the hormetic effect of growth stimulation, the physiological factor (assimilation capacity) and the biomass allocation factor are both important to consider.

The objective of this study was to first test the effects of the shoot extract of S. canadensis on the growth and biomass accumulation of maize seedlings and, second, to elucidate the possible mechanisms underlying the promotion or inhibitory effects. Specifically, we investigated how the extracts of S. canadensis affected the growth and biomass accumulation and allocation patterns of treated maize seedlings. We correlated the maize seedling shoot height to biomass allocation patterns (LAR, specific leaf area (SLA) and leaf mass ratio (LMR)) to determine whether the biomass allocation pattern could explain the promotion or inhibition observed. We also compared the photosynthetic capacity under the concentrations that either promoted or inhibited growth to determine whether the variation in photosynthetic capacity could explain the hormetic effects.

Materials and Methods

Water extract preparation

In September 2017, when S. canadensis plants started to flower, the shoots of S. canadensis plants were collected from fields and immediately transported to the laboratory, where the inflorescence was removed. The shoots were collected during this period because in this stage, S. canadensis reportedly accumulates the highest content of phenolics, which are assumed to be the major allelopathic substances in this species (BaleÞEntienë, 2015). As shoots are proven to be the most allelopathic part of S. canadensis (BaleÞEntienë, 2015), only the allelopathic effects of shoots were investigated in this study. The procedure for the water extraction of shoots followed the modified methods from Meiners (2014). The shoots were cleaned with tap water and dried at room temperature. Afterward, the shoots were cut into eight mm pieces. The water extracts were made with a ratio of one g of shoot pieces:four ml of distilled water in beakers. The extraction was performed in an incubation chamber at 30–31 °C for 24 h. Subsequently, the extraction solution was filtered through two layers of cotton and stored in a refrigerator at four °C until ready for use.

Maize seedling culture

We used maize plants as the target species, due to the frequent interaction of this crop with S. canadensis and the large yield losses associated with S. canadensis invasion (Liu et al., 2010). Commercially sold maize (Zea mays L.) seeds (var. Meiyu 8; Hainan Lvchuan Seeds Co., Ltd., Haikou, China) were germinated in nine mm Petri dishes, with 30 seeds placed in each dish. After germination for 6 days, and when the first leaf was fully open, 72 seedlings with similar size (plant height and leaf number) were transplanted into 180 ml pots containing 120 g of silica sand, with one plant in each pot. For each treatment concentration, six replicate plants were used. The maize seedlings were grown for another 4 days, and then, the first of the water extract supplement treatments was applied. The seeds were germinated and the seedlings cultured in a plant growth chamber, with a constant irradiance of 250 μmol·m2·s−1 (photosynthetically active radiation, (PAR)) above the shoots and an air temperature of 24 °C for 12 h during the day and 18 °C for 12 h at night.

Experimental design and water extract supplement treatments

To prepare the water extract concentration gradient, the original shoot water extract was diluted with distilled water. The final concentrations were as follows: 0, 0.0125, 0.025, 0.05, 0.075, 0.1, 0.125, 0.15, 0.175, 0.2, 0.225 and 0.25 g shoot equivalent/ml (hereafter referred to as g/ml). The seedlings were supplemented with the water extract from the top of the pots. Every 3 days for 9 days, each pot was treated with 100 ml of nutrient solution (1/4-strength Hoagland’s solution) and four ml of extract solution. For a total of 9 days, the water extract treatments continued. The treatments were applied under the same growth chamber conditions as described above.

Gas exchange measurement

The gas exchange measurements were performed on maize seedlings treated with three concentrations of water extract: zero (CK, control), 0.0125 g/ml (Low concentration) and 0.2 g/ml (High concentration). The measurements were conducted on the day preceding the commencement of treatments (0 day) and on the following 3 days. Net photosynthetic rates (Pn) and respiration rates (Rd) were measured on the fourth fully expanded leaf with an open-type gas exchange system (LI-6400; Li-Cor Inc., Lincoln, NE, USA). Photosynthetic light response curves were individually analyzed for the six replicated seedlings. The PAR for the light response curve was as follows: 2,500, 2,000, 1,500, 100, 500, 300, 100, 50 and 0 μmol·m2·s−1, and the flow rate was 0.5 l·min−1. The stability waiting time in the light response curve autoprogram was set as 60∼120 s. Before each measurement, the leaves were light activated for 20 min at the PAR of 2,500 μmol·m2·s−1. Light-saturated net photosynthetic rate (Pmax), apparent quantum yield (AQY) and dark respiration (Rd) were determined by using the following model (Lewis, Olszyk & Tingey, 1999): P = PmaxPPFDK + PPFD−Rd

where P is the simulated photosynthesis rate calculated with the above model, PPFD is the photosynthetic photon flux density, and K is equal to the PPFD required to produce one-half of the light-saturated photosynthetic rate.

Growth measurement

When the photosynthesis measurement was finished, the maize seedlings had developed three to four true leaves. When the photosynthesis measurement was finished, the maize seedlings had developed three to four true leaves. The maize seedlings were then harvested after the third extract application. The height of each plant was measured with a ruler, while the roots were washed with distilled water to remove any silica sand. The roots and leaves were then separated from the plants and scanned with a Microteck Scanwizard 5 (Microtek International, Inc., Xinzhu, Taiwan, China). Next, the total root length and the total leaf length and area were analyzed with the programs Winrhizo and Winfolia (Regent Instruments Inc., Quebec City, Quebec, Canada), respectively. Plant materials were then oven dried at 60 °C for 72 h, and the mass weight of the leaf, stem and root for each seedling was measured. The SLA, LMR and LAR were calculated as follow: SLA = total leaf area/total leaf weight; LMR = total leaf weight/whole plant weight; and LAR = total leaf area/whole plant weight.

Data analysis and statistics

Compared with the controlled plants (water extract concentration = 0), the change in plant height, total root length, total leaf length and area, SLA, LMR, LAR and biomass was calculated as follows: (Growth of treated plants—growth of controlled plants)/growth of controlled plants ×100%. The effects of the water extract concentration on plant growth were analyzed with one-way ANOVA (analysis of variance). The gas exchange characteristics (light-saturated Pn, Rd and AQY) were analyzed with two-way ANOVA with water extract concentration and treatment time (days) as fixed factors. Linear regression analysis of biomass based on SLA, LMR and LAR was performed with the data pairs of the plant height value and the SLA, LMR and LAR values of each individual plant. All analyses were conducted in the SPSS 16.0 statistical software package (SPSS 16.0; SPSS Inc., Chicago, IL, USA).

Results

The effect of S. canadensis water extracts on maize seedling growth and biomass accumulation followed a hormetic pattern, that is, seedling growth and biomass accumulation increased at low concentrations (0.0125–0.125 g/ml) and decreased at high concentrations (0.125–0.25 g/ml) (Figs. 1 and 2), with the most remarkable stimulatory effects at 0.0125 g/ml (Figs. 1 and 2). The effects of water extracts on plant height, total leaf length, total leaf area and total root length (P < 0.05, Table 1) were significant, whereas the effects on the biomass accumulation of leaf, stem, root and whole plant and on the resource allocation indexes SLA, LMR, LAR and root/shoot ratio were not significant (P > 0.05, Table 1).

Figure 1 Growth of the maize seedlings exposed to the different water extract concentration treatments.

(A) Plant height, (B) Total leaf length, (C) Total leaf area and (D) Total root length. 0–12 on the horizontal axis corresponds to water extract concentration: 0, 0.0125, 0.025, 0.05, 0.075, 0.1, 0.125, 0.15, 0.175, 0.2, 0.225 and 0.25 g/ml, respectively. The points in the plot refer to mean ± standard error.

Figure 2 Biomass of the maize seedling treated with different concentrations of shoot water extract from S. canadensis.

Biomass of (A) Stem, (B) Leaf, (C) Root and (D) Whole plant of the maize seedlings.

Table 1 One-way ANOVA analysis for growth and biomass accumulation of the maize seedlings treated with different concentrations of water extracts from shoots of the S. canadensis plants.

Growth	df	F	PZ		Growth	df	F	PZ	
Plant height	11.59	7.242	0.000***		Stem mass	11.59	1.763	0.081ns	
Total leaf length	11.59	2.551	0.010*		Leaf mass	11.59	1.597	0.123ns	
Total leaf area	11.59	2.204	0.026*		Root mass	11.59	1.187	0.316ns	
Total root length	11.59	4.284	0.000***		Whole plant mass	11.59	1.494	0.158ns	
LMR	11.59	1.173	0.325ns		LAR	11.59	1.502	0.155ns	
SLA	11.59	1.891	0.059ns		Root/shoot ratio	11.59	1.017	0.433ns	
Notes:

ns No significance, When P > 0.05, it is considered to be not significant.

Z P is the significance of the statistics. When P < 0.05, it is considered to be significant.

* P < 0.05

*** P < 0.001.

Hormetic-like effects were also observed on SLA and LAR (Fig. 3), but the effect of treatment concentration on these indexes was not significant (Table 1). The maximum photosynthesis rate and apparent quantum efficiency were not significantly different among the three different water extract treatments: CK (zero g/ml), low concentration (0.0125 g/ml) and high concentration (0.2 g/ml) (P > 0.05, Table 2; Fig. 4). However, those plants that received the low dose of water extract had higher respiration rates than those of the two other treatments (P < 0.05, Fig. 4). Although SLA, LAR and LMR were not significantly affected by the water extract treatments, the pattern of change in these morphological parameters was similar to that of plant shoot height. The regression analysis indicated that plant shoot height was closely associated with SLA and LAR (P < 0.01) but not with LMR (P > 0.05) (Fig. 5).

Figure 3 Biomass allocation to leaf of the maize seedlings treated with different concentrations of shoot water extract from S. canadensis.

(A) Specific leaf area (SLA), (B) Leaf mass ratio (LMR) and (C) Leaf area ratio (LAR). The points in the plot stand refer to mean ± standard error.

Table 2 Two-way ANOVA analysis for gas exchange characteristics of the maize seedlings treated with different concentrations of water extract from shoots of the S. canadensis plants.

Variation source	df	F	PZ	
Light-saturated photosynthesis rate	
 Treatment	2	0.095	0.91ns	
 Day	3	3.859	0.022*	
 Treatment × day	6	0.055	0.999ns	
Apparent quantum efficiency	
 Treatment	2	0.05	0.951ns	
 Day	3	2.653	0.072ns	
 Treatment × day	6	0.043	0.96ns	
Leaf dark respiration rate	
 Treatment	2	5.856	0.004**	
 Day	3	2.261	0.126ns	
 Treatment × day	6	0.968	0.467ns	
Notes:

CK (0.000 g/ml), Low concentration (0.0125 g/ml) and High concentration (0.2 g/ml);

ns No significance, When P > 0.05, it is considered to be not significant.

Z P is the significance of the statistics. When P < 0.05, it is considered to be significant.

* P < 0.05

** P < 0.01

Figure 4 Gas exchange characteristics of the maize seedlings treated with the three shoot water extract concentrations from S. canadensis.

CK (control, 0.000 g/ml), Low concentration (0.0125 g/ml) and High concentration (0.2 g/ml). (A) Pmax, light-saturated photosynthesis rate, (B) AQY, apparent quantum efficiency and (C) Rd, dark respiration rate. The points in the plot stand refer to mean ± standard error.

Figure 5 Regression of biomass allocation to leaf against the plant shoot of the maize seedlings treated with different concentrations of shoot water extract from S. Canadensis.

(A) Specific leaf area (SLA), (B) Leaf mass ratio (LMR) and (C) Leaf area ratio (LAR) of the maize seedlings treated with different concentrations of shoot water extract from S. canadensis.

Discussion

We observed that the water extracts of S. canadensis shoots had hormetic-like effects on the growth of maize seedlings, which is a result consistent with the findings of Sun et al. (2006) and Zhang et al. (2012), suggesting that the potential allelopathic effects of the water extracts on recipient plants were highly dependent on the concentration. In other studies, negative allelopathic effects of S. canadensis are reported (Butcko & Jensen, 2002; Abhilasha et al., 2008). The variation in the effects of water extracts on recipient plant growth may be due to the tissues studied, methods used for preparation, concentration dose of water extract applied, growth conditions and species of the recipient plant. Our study showed that identifying the potential allelopathic compounds and determining their actual concentrations in the environment are critical when regarding the allelopathic effects of S. canadensis.

We observed hormetic-like effects of the water extracts on the growth and biomass accumulation of maize seedlings. The shoot and root growth was promoted significantly by the low-dose water extract, but maize seedling biomass accumulation was not significantly promoted. The lack of significant effects on biomass accumulation was possibly due to the relatively low light availability (250 μmol·m2·s−1) and low CO2 concentration (ambient CO2 concentration, approximately 380 ppm) applied. Cedergreen & Olesen (2010) showed that the promotion effects on barley plant growth with low-dose glyphosate application were absent or much weaker at relatively low light availability or CO2 concentrations. We expected that an increase in the photosynthesis rate would explain the enhanced growth; however, no significant effects on the photosynthetic capacity of maize seedlings were observed (Fig. 3), although the respiration rate was indeed higher for the low water extract concentration of 0.0125 g/ml than that of the control and the 0.2 g/ml treatment. Similar to our results, application of low concentration microcystins, a group of cyanotoxins produced by cyanobacteria, also stimulated the growth (plant height and biomass accumulation) in rice seedlings, but did not stimulate photosynthesis rate of the plants (Liang & Wang, 2015).

Two explanations are possible for the lack of significant effects on the photosynthetic capacity. First, unlike glyphosate, the shoot water extract of S. canadensis is a mixture of diverse compounds that includes small to large molecules with differential dose-effect curves or that even cause effects in contrasting directions. Therefore, the integrated action of these compounds may lead to less pronounced effects compared with the unique effects of glyphosate (Duke et al., 2006). These compounds in S. canadensis have been categorized as phenolics, flavones and saponins (Yuan et al., 2013; BaleÞEntienë, 2015). The other explanation is that the activity of Rubisco, RuBP regeneration or the use rate of triose phosphate was not enhanced in the maize leaves due to the similar leaf N content. The leaf photosynthesis rate under ambient air conditions is most limited by Rubisco activity and leaf nitrogen content (Sinclair & Horie, 1989; Makino, 2003). Thus, the photosynthesis rate in our study was not affected because the water extract treatment may not have increased the leaf nitrogen content.

The increase in the dark respiration rate of maize seedlings at the low concentration of water extract, which is similar to that in barley plants treated with a low dose of glyphosate (Cedergreen & Olesen, 2010), may explain the increase in maize plant height, leaf area and length and root length. In hormesis, the increase in root and shoot growth is interpreted as an adaptive mechanism of escape from stressful conditions (Duke et al., 2006). The increase in respiration rate indicated increased metabolic activity in response to the toxic water extract, which may enable the recipient plant to activate detoxification, inactivation or compartmentalization processes (Cedergreen & Olesen, 2010). In addition, we observed a pattern of allocating more resources toward aboveground growth at low water extract concentrations, which could be a strategy to escape from the harmful underground conditions, even though root growth was also promoted at low water extract concentrations. Other environmental factors, such as mineral nutrient supplements, play important roles in growth stimulation by increasing the aboveground biomass allocation parameters, such as SLA and LAR (Poorter & Nagel, 2000). The increases in SLA and LAR suggested greater allocation to shoot and leaf growth, which can increase the photosynthetic area relative to other nonphotosynthetic organs. The concentration range that stimulated shoot growth overlapped with the range that stimulated SLA and LAR (Figs. 1–3). Furthermore, the positive correlations between SLA and LAR and plant shoot height suggested that the promotion of growth with the low-dose water extract was due more to the increase in assimilation area than to the increase in assimilation capacity (per unit leaf area). The increase in SLA may be explained by either a moderate increase in reactive oxygen species (Prithiviraj et al., 2007) or an inhibition of lignin synthesis (Duke et al., 2006), which both occur at low-dose treatments. In another study on the hormetic-like effects of S. canadensis extracts, the ability of the recipient plant to cope with stress, as indicated by the activities of Superoxide dismutase and other enzymes, was stimulated at low extract concentrations but was inhibited at high concentrations (Zhang et al., 2012), suggesting that the mechanisms also act at the physiological level.

Although whether a stimulatory effect occurs in fields where S. canadensis invades is not clear, fast growth together with high SLA and LAR could have significant ecological consequences for those plants that are affected by low-dose phytotoxins because these plants with relatively fast growth rates may have increased vulnerability to mechanical stress or herbivory (Belz, Cedergreen & Duke, 2011). These secondary consequences should also be considered when explaining crop yield loss caused by exotic plant invasion.

In summary, the water extract of S. canadensis had significant effects on maize seedling growth, suggesting that the interactions of S. canadensis with crops or native species in fields could be affected. The effects were highly dependent on the concentration; the stimulated growth caused by the low-concentration water extract of S. canadensis shoots on maize seedlings growth could be explained mostly by the biomass allocation patterns (leaf SLA and LAR) but not by the gas exchange capacity. Therefore, the investigation of the concentration of allelopathic compounds of S. canadensis in the field is critical to the study of their possible effects on native species.

Supplemental Information

Supplemental Information 1 Supplemental S1.

Click here for additional data file.

We thank the editors and reviewers for their constructive comments and suggestions. We thank Prof. Xin Chen for her kind help in revision of the manuscript.

Additional Information and Declarations

Competing Interests

Author Contributions

Data Availability

The authors declare that they have no competing interests.

Xiao qi Ye conceived and designed the experiments, performed the experiments, analyzed the data, contributed reagents/materials/analysis tools, prepared figures and/or tables, authored or reviewed drafts of the paper, correspondence with the editor.

Jin liu Meng performed the experiments, analyzed the data, contributed reagents/materials/analysis tools.

Ming Wu conceived and designed the experiments, prepared figures and/or tables, approved the final draft.

The following information was supplied regarding data availability:

The raw measurements are available in File S1.

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
