# Peer review of "The effects of Solidago canadensis water extracts on maize seedling growth in association with the biomass allocation pattern"

_PeerJ, doi:10.7717/peerj.6564_

## Round 0.1 · original submission · Major Revisions

The paper has been reviewed by three reviewers. Reviewer 1 was suggesting rejection of the manuscript because of the lack of novelty. Although innovation is not a criterion for the acceptance in PeerJ I would strongly encourage the authors to better highlight new points in their work.

I reread the paper and found a number of sloppily written places in the methods (description of photosynthesis measurements is not clear how long leaves were exposed to light, unit of irradiance was wrong) The extract is not well characterized (It would be good to have some idea how much Solidago dry mass is in the substance etc).

The graphs have ordinal units not extract concentrations as x-axis values.

The language should be also revised.

Reviewer 1 ·

Basic reporting

The manuscript is good in English writing. It is easy to understand what the authors want to say. Overall, the content and language of the paper is good overall.

Experimental design

The experiment methods used in this manuscript were right, but very conventional and simple. The research results are not significantly innovative in related field.

Validity of the findings

This manuscript is just about the allelopathic effects of a common alien invasive plant in China on the growth of a crop. The result is not much value to improve our understanding of the impacts of plant invasions. It is just a common subject, I think not much readers will be interested in it.

Additional comments

No more comments

Reviewer 2 ·

Basic reporting

I think that the study titled "The effects of Solidago canadensis water extracts on maize seedling growth in association with biomass allocation patterns" has valuable data about the phytotoxic effects of S. canadensis not only on plant growth but also on leaf characteristics and photosynthetic activity. However, the introduction need to be simplified since some parts are confuse and difficult to follow. The aims need to be clarify as well as some parts of the material and methods.
I have suggested small modifications in the results section. In discussion, some results need better explanations in the basis of theory concepts or the results found in literature

Experimental design

I think that the experimental design is fine and well explained in general

Validity of the findings

Althought I have made some suggestions in the results, in general they are well presented and supported by statistical analysis.

Conclusions need to be reviewed

Additional comments

I have added specific comments to the manuscript in the word file

Annotated reviews are not available for download in order to protect the identity of reviewers who chose to remain anonymous.

Reviewer 3 ·

Basic reporting

Professional English language is used throughout the manuscript.

The background is not clear enough, the authors focus mostly on hormetic effects of S. canadensis for most of the Introduction part. However, the very first lines of Introduction state a negative effect of S. canadensis on native plants.

Relevant literature was used for this article.

Article is well structured.

Experimental design

This manuscript contains original, primary research.

Aim: I don't understand why maize was used as a donor-species, especially for hormetic effects, since, as authors prove, this species is mostly inhibited by Solidago.

The research question is properly stated.

ln. 105 according to my calculations there were only 6 seedlings/repetitions per treatment. Did you repeat the experiment to get more robust results?
ln. 99 - could you cite a proper literature for extracting in a refrigerator in 4 C degrees?
ln. 100 - what does the expression "2 layers of cotton" mean?
ln. 113 - what was the extract diluted by? distilled water?
ln. 115 Why do you express the units in mg/ml, since both extract and dilutor are liquids, probably the same kind of liquid?
ln 118 were the extracts and nutrient solution applied from the top or bottom of the pot?

Validity of the findings

The results part is written in a very compacted way. The data is presented in a robust way in the figures. Statistical data is also clearly presented.
ln. 176-the expression "seem to have" is not precise, especially when you perform statistical analyses. Please correct.
ln. 191-192. The authors did not measure the Relative growth rate, still they discuss very evidently that "RGR was promoted". Vide - this conclusion is not supported by your results.
The other parts of discussion are written in a good way
Conclusion is sufficient enough, and in relation to the research hypothesis.

Additional comments

In general the manuscript should be more clear, especially in the methodological part.

---

## Round 0.2 · Minor Revisions

The revised version was reviewed by one reviewer. In addition to her comments I found a number of small mistakes in language and structure that should be addressed. There are probably more issues in the language that have escaped me.

line 31 via alteration of biomass allocation patternS'
line 39 (photosynthetic light response curve)
line 71 needs a reference
line 94 the sentence is not correct.... can BE due to A larger
line 96 the sentence is quite cumbersome please rephrase
line 114 Itjust came to my mind that it would be nice to know more about the S candiensins (when was it collected (which day of year) what was the phenological stage (already flowering or not yet)etc...
line 124 if would be nice to document the variety of the maize
please give more ideas on the materiasls and methods. How long did the experiment last? at which days the applications were done etc....
line 217 I wonder if the comparison of glyphosphate and S canadiensis extract is solid. There are several alelopathic pathways. Why do you think that Glyphosphate and S canadienses are similar?
line 253 I do not understand the sentence

Reviewer 3 ·

Basic reporting

The English language of this manuscript should be corrected.

Literature is sufficient.

Article is properly structured. Figures and tables are prepared in a proper way, easy to read.

I don't understand why the part Introduction is finished with a conclusion of these studies.

The conclusion part should be placed at the end of the paper. I suppose ln 97, where the Authors write about what the "suggest" in this paper is actually a research hypothesis and this should be clearly stated.

Experimental design

The research fits within the aims and scope of the journal.

Research question should be more clear (please see the comment above)

Investigation performed correctly.

Methods are described fairly. In which growth stage the maize was collected for the measurements?

Validity of the findings

Results are described very briefly. The results are not very novel, but do bring some new findings to the field. Data is statistically sound. Conclusions are linked with the original question and research results.

---

## Round 0.3 · accepted · Accept

I greatly appreciate the language editing work for the paper. I apologize for the slow treatment in this stage of the process. There was a number of deadlines in January.

#